# Navigating Professional Identity and Cultural Expectations: A Phenomenological Study of Female Saudi Nurses’ Experiences in Mixed-Gender Healthcare Settings

**DOI:** 10.3390/healthcare13233042

**Published:** 2025-11-25

**Authors:** Waleed M. Alshehri, Wjdan A. Almutairi, Thurayya Eid, Shorok H. Alahmedi, Safiya Salem Bakarman, Ashwaq A. Almutairi, Abdulaziz M. Alodhailah

**Affiliations:** 1Department of Medical-Surgical Nursing, College of Nursing, King Saud University, Riyadh 11451, Saudi Arabia; 2College of Nursing, King Saud Bin Abdulaziz University for Health Sciences (KSAU-HS), Riyadh 11481, Saudi Arabia; 3King Abdullah International Medical Research Center, Riyadh 11481, Saudi Arabia; 4Ministry of National Guard Health Affairs, Riyadh 11426, Saudi Arabia; 5Department of Nursing Management and Education, College of Nursing, Princess Nourah bint Abdulrahman University, Riyadh 11671, Saudi Arabia; 6Department of Community and Mental Health Nursing, College of Nursing, King Saud University, Riyadh 11495, Saudi Arabia; 7Monash Nursing and Midwifery, Monash University, Melbourne, VIC 3800, Australia

**Keywords:** professional identity, gender dynamics, cultural expectations, Saudi nurses, mixed-gender healthcare, phenomenology

## Abstract

Purpose: This study aimed to explore the lived experiences of Saudi female nurses working in mixed-gender healthcare environments and understand how they navigate professional identity while managing cultural expectations in Saudi Arabia’s evolving healthcare landscape. Methods: A descriptive phenomenological qualitative study grounded in symbolic interactionism was conducted using in-depth semi-structured interviews with 20 Saudi female nurses working in mixed-gender healthcare settings in Riyadh. Interviews were conducted in Arabic and systematically translated using forward–backward translation protocols. Data were analyzed using Colaizzi’s phenomenological analysis framework to identify essential themes and meanings. Trustworthiness was established through credibility, dependability, confirmability, and transferability strategies, including member checking with 6 participants, peer debriefing, and comprehensive audit trails. Results: Four major themes emerged: (1) Reconciling Traditional Values with Professional Duties, (2) Negotiating Gender Dynamics in Clinical Practice, (3) Developing Professional Identity Amid Cultural Tensions, and (4) Organizational Support and Environmental Adaptation. Participants demonstrated remarkable resilience in balancing cultural expectations with professional responsibilities while developing sophisticated coping strategies. Conclusions: Saudi female nurses actively construct their professional identities while navigating complex cultural landscapes. The study reveals the need for organizational policies that support cultural sensitivity while promoting professional growth and gender equality in healthcare settings. These findings may inform healthcare workforce development in other Islamic and culturally transitioning contexts.

## 1. Introduction

Saudi Arabia’s healthcare system transformation under Vision 2030 has fundamentally restructured the professional landscape for healthcare workers, creating unprecedented opportunities and challenges, particularly for women in nursing [1,2]. This transformation represents more than policy reform; it constitutes a cultural paradigm shift that challenges traditional gender roles while maintaining Islamic values and Saudi cultural identity [3,4]. Within this evolving context, the introduction of mixed-gender healthcare environments marks a significant departure from historically gender-segregated practice settings, positioning female nurses at the center of this cultural and professional transformation [5].

Female nurses currently comprise approximately 65% of Saudi Arabia’s nursing workforce, representing both the largest professional group within healthcare and the most visible symbol of women’s expanding role in the Kingdom’s economic diversification strategy [2]. These professionals find themselves navigating a complex intersection where professional advancement opportunities converge with deeply rooted cultural expectations, family obligations, and evolving societal norms [6]. The complexity of this navigation process extends beyond individual adaptation to encompass broader questions about cultural preservation, professional identity formation, and the integration of Islamic values within contemporary healthcare practice.

Professional identity in nursing encompasses the adopted values, beliefs, assumptions, and behavioral patterns that define one’s professional self-concept and guide practice decisions [7]. However, for Saudi female nurses, this identity formation process occurs within a distinctive cultural matrix that includes Islamic principles, tribal traditions, family expectations, and rapidly evolving governmental policies supporting women’s workforce participation, Understanding how these nurses construct and maintain professional identities while honoring cultural commitments has become crucial for supporting their career development, ensuring sustainable healthcare workforce growth, and maintaining quality patient care delivery [8].

### 1.1. Theoretical Framework

This study employs symbolic interactionism as its theoretical foundation, recognizing that individuals actively construct meaning through social interactions and interpret their experiences within specific cultural contexts [8]. Symbolic interactionism posits that professional identity is not fixed but continuously negotiated through interactions with others, patients, colleagues, families, and society, making it particularly relevant for understanding how Saudi female nurses navigate multiple, sometimes competing, role expectations in mixed-gender healthcare environments [6]. This theoretical lens informed our research questions, interview guide development, and analytic approach by focusing on meaning-making processes, role negotiation strategies, and identity construction through social interaction.

While we employ symbolic interactionism as our guiding theoretical framework, we have adopted Husserl’s descriptive phenomenology and Colaizzi’s analysis method to capture the essence of lived experiences [9]. We acknowledge this epistemological pairing requires justification: symbolic interactionism emphasizes meaning construction through social interaction (a constructionist stance), whereas Husserlian [9] phenomenology seeks the essential structures of experience. We integrate these approaches by using interactionism to frame what we study (how nurses construct professional identity through cultural negotiation) while employing descriptive phenomenology to understand how they experience this process. The Colaizzi [9,10] method enables systematic identification of meaning units grounded in participants’ own words, which we then interpret through an interactionist lens that recognizes identity as actively constructed rather than passively received. This dual framework acknowledges both the subjective essence of nurses’ experiences and the social processes through which those experiences are shaped.

### 1.2. Research Significance and Knowledge Gap

Despite extensive global literature addressing nursing professional identity development, limited research has examined the unique experiences of nurses working within Islamic cultural contexts, particularly during periods of rapid social transformation [10]. Saudi Arabia’s distinctive combination of Islamic governance, tribal cultural traditions, and accelerated modernization creates a unique context for understanding professional identity formation among female healthcare workers [8].

Previous studies examining nursing professional identity have predominantly focused on Western healthcare contexts where cultural homogeneity and established gender equality norms create different identity formation challenges. The gap in understanding how nurses from Islamic cultural backgrounds navigate professional development within mixed-gender environments represents a significant limitation in current nursing knowledge, particularly as healthcare systems globally become increasingly diverse and culturally complex [11].

Furthermore, existing research has often approached cultural considerations as barriers to professional development rather than examining how cultural values might enhance or inform professional practice [12]. This study addresses these knowledge gaps by providing nuanced insights into how Saudi female nurses actively integrate cultural identity with professional competence, contributing to broader understanding of cultural diversity’s role in healthcare workforce development.

## 2. Materials and Methods

This study employed Husserl’s descriptive phenomenological approach to explore the essence of lived experiences among Saudi female nurses working in mixed-gender healthcare settings [13]. Descriptive phenomenology focuses on understanding phenomena as experienced by participants, seeking to uncover the essential structures of experience while bracketing researchers’ preconceptions [14]. As detailed above, we integrated this with symbolic interactionism to frame identity as socially constructed through ongoing negotiation.

### 2.1. Researcher Characteristics and Reflexivity

Following COREQ guidelines [15] we describe the research team’s background. The principal investigator is a male Saudi nurse educator with more than 15 years of clinical and academic experience and a PhD in nursing. The co-investigators are female Saudi nurse researchers with expertise in qualitative methods, cultural competence, and nursing education. A senior methodological expert oversaw the study’s rigor. The team contributed insider knowledge of Saudi healthcare while managing potential biases through regular reflexive discussions.

Before data collection, the team conducted bracketing exercises to identify and set aside assumptions about gender roles, professional identity, and cultural tensions in Saudi nursing. Reflexive journals were maintained and revisited throughout analysis to ensure findings were grounded in the data. The interviewers positioned themselves as fellow Saudi nurses seeking to understand participants’ experiences, which supported openness, though the gender of the interviewers may have influenced some responses.

No team member had prior relationships with participants. Participation was voluntary, and all participants were informed that declining would not affect their employment or professional standing.

### 2.2. Setting and Contextual Framework

The study was conducted with participants working across multiple healthcare institutions in Riyadh, Saudi Arabia’s capital city, which hosts the largest healthcare facilities, including governmental tertiary care hospitals, private hospitals, and specialized healthcare facilities. All participants were working in mixed-gender healthcare facilities where both male and female healthcare professionals and patients were present.

This multi-site selection strategy ensured representation across different organizational cultures, patient populations, and healthcare delivery models. The diversity of institutional contexts strengthened the study’s ability to capture varied experiences while identifying common themes across different settings.

### 2.3. Participant Selection and Characteristics

Twenty Saudi female nurses were recruited through purposive sampling to capture diverse perspectives across clinical specialties, career stages, and educational backgrounds (Table 1). Sample size was guided by information power [16], which emphasizes the relevance and richness of data over participant numbers. Our sample provided sufficient information power due to: (1) a narrow, well-defined study aim; (2) a highly specific population of Saudi female nurses in mixed-gender settings; (3) theory-driven, in-depth interviews; (4) case-oriented analysis using Colaizzi’s method; and (5) cross-case pattern identification.

Recruitment used social media posts, professional nursing networks, and snowball referrals. Of 24 nurses expressing interest, 22 met inclusion criteria, and 20 consented. Two declined due to scheduling conflicts, and no participants withdrew after consent.

Inclusion criteria comprised Saudi nationality, female gender, current registration with Saudi Commission for Health Specialties license, minimum two years of experience in mixed-gender healthcare facility, and voluntary willingness to share experiences openly. Exclusion criteria eliminated nurses working exclusively in gender-segregated units, student nurses or interns, and non-Saudi nationals whose cultural experiences might differ significantly from the target population.

### 2.4. Data Collection Procedures

Individual, in-depth, semi-structured interviews were conducted in Arabic by the researchers, lasting 25–60 min (mean = 42) depending on participant elaboration. Interviews were held in person or online based on participants’ preferences and schedules, with all conducted one-on-one to ensure comfort and openness. No third parties were present except in two cases where participants requested a female colleague nearby for cultural comfort.

All interviews were audio-recorded with explicit participant consent and conducted in private settings to maintain confidentiality and encourage honest reflection.

The interview guide (see Appendix A) was developed based on phenomenological inquiry principles, extensive literature review, symbolic interactionist theory, and consultation with three cultural competency experts familiar with Saudi healthcare contexts. The guide included opening questions about daily work experiences, main questions exploring cultural–professional tensions and identity negotiation strategies, and closing questions about organizational support and future aspirations. Pilot testing with three participants (excluded from final analysis) refined question wording, sequence, cultural sensitivity, and probe effectiveness. Example questions included:“Can you describe a typical day for you as a nurse in a mixed-gender healthcare environment?”“How do you experience the intersection between your cultural identity as a Saudi woman and your professional role as a nurse?”“Tell me about a time when you felt tension between cultural expectations and professional responsibilities. How did you navigate that situation?”“What strategies have you developed to maintain your Islamic values while working with male colleagues and patients?”“How has your sense of professional identity evolved since you began working in mixed-gender settings?”

The interviews focused on understanding Saudi nurses’ everyday experiences of working in mixed-gender environments. Participants spoke about how cultural values and expectations shape their professional lives and the tensions they sometimes face between their roles as nurses and the expectations of family and society. They shared stories about the unique challenges of working alongside male colleagues, as well as the strategies they use to adapt and manage these cultural and professional pressures. The discussions also touched on the kind of support they receive from their workplaces, how the work environment affects their confidence and comfort, and how these experiences influence their sense of professional identity and hopes for the future.

### 2.5. Translation Procedures

As all interviews were conducted in Arabic, rigorous translation procedures were followed to maintain data integrity and preserve cultural and religious nuances, based on established guidelines for cross-language qualitative research [17,18]. The workflow included:Verbatim transcription: Audio recordings were transcribed in Arabic by a professional familiar with medical terminology and Saudi dialects; transcripts were reviewed against recordings for accuracy.Forward translation: Two bilingual Saudi nurse researchers independently translated transcripts into English, preserving idiomatic and cultural meanings.Comparison and consensus: Translators discussed discrepancies and agreed on optimal English phrasing.Back-translation verification: A third bilingual researcher back-translated 10% of the text to confirm meaning equivalence.Cultural and religious expression handling: Islamic terms (e.g., hijab, Ramadan, worship as caring) were retained in Arabic with English explanations; semantic equivalence was prioritized over literal translation.Participant review: Six participants confirmed that their intended meanings were accurately preserved during member checking.

This process ensured that the richness, nuance, and cultural specificity of participants’ narratives were maintained throughout translation and analysis.

### 2.6. Data Analysis Framework

Data analysis followed [19] systematic seven-step phenomenological analysis method, chosen for its rigorous approach to identifying essential meanings while maintaining fidelity to participants’ lived experiences. This analytical framework progresses from individual statements to broader thematic patterns, ensuring both depth and breadth in understanding the phenomenon.

The seven analytical steps comprised: (1) Reading all participant transcripts thoroughly to develop overall impressions; (2) Extracting significant statements directly related to the research phenomenon; (3) Formulating meaning statements for each significant statement; (4) Organizing formulated meanings into coherent theme clusters; (5) Integrating theme clusters into comprehensive phenomenon descriptions; (6) Identifying fundamental structural elements of the experience; and (7) Conducting member checking with selected participants to validate interpretive accuracy as per Colaizzi’s original method.

Two researchers independently coded all transcripts, identifying significant statements and formulating meanings. Initial agreement was reached on 85% of statements, with disagreements resolved through discussion, reference to the original transcripts, and consultation with a third researcher when needed. Theme clusters were developed by the full team in three extended meetings, grouping and refining meanings until coherent themes emerged. This collaborative approach enhanced analytical rigor while ensuring cultural sensitivity in interpreting participants’ experiences within Saudi contexts.

### 2.7. Ethical Considerations and Trustworthiness

Ethical approval was secured from King Saud University’s Institutional Review Board (KSU-HE-25-952), with all procedures conducted according to Helsinki Declaration principles and Saudi research ethics guidelines. Written informed consent was obtained from all participants, clearly explaining study purposes, voluntary participation, confidentiality protections, and withdrawal rights without consequences.

Trustworthiness was ensured using [20] criteria. Credibility was established through prolonged engagement (each transcript read ≥5 times by two researchers), member checking with six participants who confirmed theme accuracy, and bi-weekly peer debriefing with external qualitative experts. Dependability was supported by a detailed audit trail (reflexive journals, coding decisions, meeting notes, theme evolution charts) and code-recode reliability (15% of transcripts recoded, 90% consistency). Confirmability was maintained via reflexive bracketing, data-grounded interpretations requiring ≥75% participant representation across ≥3 clinical settings, and negative case analysis. Transferability was addressed through thick description of participant demographics, context, and narratives, and explicit geographic, cultural, and temporal boundaries.

Cultural sensitivity was ensured through ongoing consultation with cultural experts and careful handling of Islamic and cultural references. Participant confidentiality was protected using pseudonyms and removal of identifying details.

## 3. Results

Four major themes emerged from the phenomenological analysis (Table 2), revealing the complex and sophisticated strategies employed by Saudi female nurses to navigate professional identity development within mixed-gender healthcare environments. These themes demonstrate how participants actively construct integrated professional identities that honor both cultural values and contemporary nursing practice requirements.

To illustrate the analytical process and ensure transparency, Table 3 provides a detailed traceability example showing how participants’ significant statements were transformed into formulated meanings, grouped into theme clusters, and ultimately mapped to final themes. This demonstrates how Colaizzi’s method systematically progressed from participants’ own words (Step 2) to researcher-formulated meanings (Step 3), clustered patterns (Step 4), and overarching themes (Step 5). Steps 6–7 involved synthesizing these themes into a comprehensive description of the phenomenon and conducting member checking, as detailed in the trustworthiness section.

### 3.1. Theme 1: Reconciling Traditional Values with Professional Duties

This foundational theme captures the sophisticated integration processes through which participants harmonize deeply held Islamic values and Saudi cultural traditions with contemporary professional nursing responsibilities. Rather than experiencing these as competing demands, participants developed creative synthesis approaches that enhanced both cultural authenticity and professional competence.

#### 3.1.1. Cultural Value Preservation Within Professional Contexts

Participants consistently emphasized their commitment to maintaining Islamic identity and cultural practices while excelling in mixed-gender professional environments. This preservation process required active negotiation rather than passive accommodation, with nurses developing specific strategies for honoring religious obligations within healthcare settings. A nurse with eight years of experience articulated this integration:

“Working in a mixed environment doesn’t mean abandoning who I am as a Saudi woman. I maintain my Islamic values, I observe proper hijab, I interact with male colleagues professionally and respectfully, and I ensure my behavior reflects our cultural principles. But I also provide the best nursing care possible, regardless of patient gender. These aren’t contradictory; they’re complementary aspects of who I am as a professional.” (P1)

The challenge of maintaining religious practices while meeting clinical demands emerged as a significant concern requiring organizational understanding and personal creativity. Participants described developing time management strategies, seeking prayer accommodations, and educating colleagues about religious obligations. A critical care nurse with six years of experience shared her family negotiation process:

“My family was initially concerned about me working with male healthcare providers and patients. I had to show them that I could maintain my religious obligations while excelling in my profession. I pray at work, I fast during Ramadan while caring for patients, and I demonstrate that Islamic values enhance rather than hinder my nursing practice. Now my family sees nursing as an extension of Islamic caring principles.” (P2)

Not all participants experienced smooth family acceptance. One participant noted: 

“My extended family still questions my career choice. They worry about my reputation and marriage prospects. This ongoing tension requires constant emotional energy”.(P5)

This counter-example illustrates the variability in family support across participants’

#### 3.1.2. Professional Duty Integration Through Religious Framework

Participants reframed nursing practice within Islamic humanitarian principles, transforming potential cultural conflicts into sources of professional strength and motivation. This reframing process involved interpreting patient care as worship, viewing professional competence as religious obligation, and understanding mixed-gender interactions as necessary for serving humanity. An emergency nurse with 10 years of experience explained this theological integration:

“There’s a misconception that working in mixed environments compromises our Islamic identity. Actually, I feel my faith strengthens my nursing. When I care for patients, I see it as an act of worship serving humanity as Allah commanded. The Quran teaches that saving one life is like saving all humanity. My hijab becomes a symbol of professional dignity, not limitation. Islam actually calls us to excellence in all endeavors, including healthcare.” (P3)

This religious reframing proved particularly powerful during challenging clinical situations where cultural boundaries might otherwise create hesitation. A pediatric nurse with five years of experience described how theological understanding guided her practice:

“Emergency situations don’t recognize gender boundaries. When a child is coding, I don’t think about who’s in the room male doctors, fathers, brothers. My focus is entirely on saving that life, which is the highest Islamic principle. My family has come to understand that sometimes professional duty exceeds social conventions because it serves higher Islamic purposes.” (P4)

### 3.2. Theme 2: Negotiating Gender Dynamics in Clinical Practice

This theme explores the complex interpersonal and professional negotiations required for effective clinical practice within mixed-gender healthcare teams, revealing both challenges and opportunities inherent in these working relationships.

#### 3.2.1. Professional Boundary Management and Credibility Establishment

Participants developed sophisticated boundary management strategies that maintained cultural appropriateness while establishing professional authority and clinical credibility. These strategies required continuous calibration based on specific situations, colleague relationships, and patient needs. A medical-surgical nurse with seven years of experience described her approach:

“Working with male colleagues requires clear professional boundaries that demonstrate competence while respecting cultural norms. I maintain appropriate eye contact during clinical discussions but avoid unnecessary physical contact. I participate fully in medical rounds, voice clinical opinions confidently, and ensure my interactions remain strictly professional. It’s about earning respect through competence while maintaining cultural appropriateness. Respect comes from clinical expertise, not from compromising cultural values.” (P6)

The process of establishing professional credibility often required overcoming initial gender-based assumptions about competence and authority. This challenge proved particularly pronounced for younger nurses or those working in traditionally male-dominated specialties. A critical care nurse with 11 years of experience reflected on this credibility-building process:

“Some male healthcare providers initially questioned my clinical judgment, assuming I was less capable because I’m a Saudi woman wearing hijab. They would double-check my assessments or bypass me in clinical discussions. I had to prove myself repeatedly through clinical excellence demonstrating high assessment skills, anticipating complications, advocating effectively for patients. Now, these same healthcare providers seek my input and trust my clinical judgment because they’ve seen my expertise. Respect isn’t given, it’s earned through consistent professional excellence and patience.” (P7)

Night shifts and emergency situations often posed added challenges for maintaining professional boundaries while ensuring quality care. As one emergency nurse with four years of experience explained:

“Gender dynamics can feel more complicated at night when fewer staff are around. Sometimes male colleagues make inappropriate comments or assumptions about social interaction. I’ve learned to stay professional, keep conversations focused on work, and report issues if needed. Protecting my integrity and cultural values is very important to me.” (P8)

#### 3.2.2. Patient Care Across Gender Lines and Family Dynamics

Providing care to male patients required nurses to navigate cultural expectations while maintaining professionalism and patient comfort. Many described developing confidence through communication, education, and experience. A medical-surgical nurse with six years of experience shared:

“At first, interacting with male patients felt a bit challenging because I wanted to ensure respect and comfort for everyone involved. Over time, I focused on clear communication, professionalism, and empathy, explaining procedures, maintaining privacy, and ensuring patients felt safe and respected. When patients sense genuine care and confidence, gender becomes less of a concern.” (P9)

Family expectations sometimes added another layer of complexity, as relatives occasionally questioned the role of female nurses in caring for male family members. An intensive care nurse with eight years of experience explained:

“Families can be protective, especially with older patients. Sometimes they’re unsure at first, but I stay calm and focus on delivering attentive, high-quality care. Once they see the results, their trust grows quickly. Professionalism and kindness usually overcome hesitation.” (P10)

### 3.3. Theme 3: Developing Professional Identity Amid Cultural Tensions

This theme examines the dynamic, ongoing process through which participants construct coherent professional identities that integrate multiple role expectations, cultural values, and professional competencies into unified self-concepts.

#### 3.3.1. Identity Construction as Integration Rather than Choice

Participants consistently rejected binary conceptualizations that positioned cultural identity and professional identity as competing alternatives. Instead, they described active integration processes that created new, hybrid identity forms honoring both cultural authenticity and professional excellence. A critical care nurse with nine years of experience articulated this integration philosophy:

“Becoming a professional nurse in Saudi Arabia means constantly negotiating who I am, but not choosing between identities. At work, I’m confident, assertive, make life-saving decisions, and lead healthcare teams. At home, I maintain cultural traditions, show family respect, and fulfill traditional responsibilities. I’ve learned to integrate both aspects, my professional confidence actually makes me a stronger Saudi woman, and my cultural grounding makes me a more compassionate nurse. These identities reinforce rather than compete with each other.” (P11)

The evolution from cultural accommodation to professional assertiveness emerged as a developmental process requiring time, experience, and supportive environments. Participants described gradual transitions from attempting to minimize their professional presence to confidently claiming professional space and authority. An emergency nurse with seven years of experience shared her identity evolution:

“My nursing identity evolved gradually through stages. Initially, I tried to minimize my presence, avoid conflicts, be invisible professionally to avoid cultural criticism. But critical care nursing demands presence, authority, quick decision-making, and team leadership. I had to find my professional voice, assert clinical judgment confidently, and claim my rightful space in healthcare teams while remaining culturally respectful. This wasn’t abandoning culture, it was discovering that Islamic values actually support professional excellence and leadership when properly understood.” (P12)

The concept of creating “new integrated identities” emerged as participants rejected traditional either-or frameworks in favor of both-and approaches that expanded rather than limited their self-concepts. A pediatric nurse with five years of experience expressed this philosophy:

“People sometimes ask if I feel conflicted between being a professional and being Saudi, as if I must choose one identity over the other. But I don’t see it as conflict. I see it as integration and expansion. My cultural background gives me empathy, deep understanding of family dynamics, and communication skills that enhance my nursing practice. My professional training gives me confidence, decision-making skills, and leadership abilities that make me a stronger Saudi woman. I’m not choosing between identities; I’m creating a new integrated identity that’s both authentically Saudi and professionally excellent.” (P13)

#### 3.3.2. Role Model Development and Intergenerational Influence

Participants recognized their pioneering role in demonstrating successful professional integration for younger Saudi women entering healthcare fields. This recognition created additional responsibility but also served as motivation for professional excellence and cultural authenticity. A medical-surgical nurse with 12 years of experience reflected on this mentorship role:

“I realize I’m a pioneer for younger Saudi women entering nursing and other healthcare professions. They watch carefully how I balance professionalism with cultural values, how I handle challenges, how I maintain family relationships while pursuing career advancement. I try to demonstrate that we can be excellent nurses and healthcare leaders without compromising our Saudi identity or Islamic values. It’s a significant responsibility that I take seriously because I’m helping create pathways for the next generation.” (P14)

Mentorship relationships with younger nurses provided opportunities for sharing integration strategies while learning from newer perspectives on cultural adaptation. A critical care nurse with eight years of experience described her mentoring approach:

“Mentoring younger Saudi nurses entering mixed-gender healthcare is crucial for our profession’s future. I share strategies for managing family expectations, dealing with workplace challenges, building professional confidence, and maintaining cultural authenticity. But I also learn from them, they often have fresh perspectives on cultural adaptation and professional integration. Together, we’re creating a new generation of Saudi nurses who are both professionally competent and culturally grounded.” (P15)

### 3.4. Theme 4: Organizational Support and Environmental Adaptation

This theme explores how organizational policies, leadership approaches, colleague relationships, and environmental factors either facilitate or hinder participants’ successful navigation of cultural and professional integration challenges.

#### 3.4.1. Organizational Policy Impact on Cultural Integration

Participants emphasized the critical importance of organizational policies that demonstrate understanding and accommodation of cultural needs while promoting professional development and career advancement. These policies created environments where Saudi female nurses could thrive professionally without sacrificing cultural authenticity. An emergency nurse with six years of experience explained the impact of supportive policies:

“Our hospital’s policies around prayer time accommodation, hijab-appropriate uniforms, and culturally sensitive scheduling show genuine respect for our cultural needs. They provide designated prayer spaces, schedule flexibility during Ramadan, and ensure appropriate gender considerations for overnight shifts when possible. When organizations demonstrate this level of cultural understanding, we feel genuinely valued and can focus entirely on providing excellent patient care. It’s not about special treatment, it’s about inclusive policies that recognize and support our dual identities as Saudi women and healthcare professionals.” (P16)

Professional development opportunities that acknowledged cultural contexts while building leadership skills proved particularly valuable for career advancement. An intensive care nurse with 10 years of experience described transformative training experiences:

“Our hospital’s leadership development programs were specifically designed to help Saudi women develop confidence in mixed-gender professional teams. They provided communication skills training, conflict resolution techniques, and cultural competency education for all staff members. These programs created an environment where Saudi women could develop and demonstrate leadership capabilities while maintaining cultural identity. I learned to lead healthcare teams confidently while respecting cultural boundaries and Islamic principles.” (P17)

Flexible scheduling policies that accommodated family obligations and cultural requirements reduced stress and enabled full professional engagement. A medical-surgical nurse with seven years of experience appreciated this organizational flexibility:

“Flexible scheduling policies that consider family obligations and cultural requirements make an enormous difference in our ability to succeed professionally. During important family events, religious holidays, or children’s school activities, supervisors work with us to adjust schedules appropriately. This flexibility reduces the stress of managing competing demands and allows us to fully commit to professional responsibilities when at work, knowing that our cultural and family obligations are also respected and supported.” (P18)

#### 3.4.2. Colleague Relationships and Cross-Cultural Support Networks

The development of supportive relationships with both Saudi and international colleagues emerged as crucial for successful adaptation to mixed-gender healthcare environments. These relationships provided emotional support, practical advice, and professional development opportunities. A pediatric nurse with four years of experience described the evolution of colleague relationships:

“My international colleagues initially didn’t understand cultural considerations, why I couldn’t attend certain social events, why family approval mattered for schedule changes, why I needed specific accommodations for religious practices. But over time, through patient education and demonstration, they’ve become supportive allies who actively help create inclusive work environments that respect cultural diversity. Now they advocate for cultural accommodations and celebrate our professional achievements while learning about Saudi culture and Islamic values.” (P19)

Peer support networks among Saudi female nurses provided essential emotional support and practical strategy sharing for navigating cultural and professional challenges. A critical care nurse with nine years of experience emphasized the importance of this peer support:

“Having other Saudi female nurses as colleagues provides crucial emotional support that others cannot fully provide. We understand each other’s unique challenges, share effective coping strategies, celebrate professional achievements together, and provide encouragement during difficult periods. This peer support network is absolutely essential for successfully navigating the complex pressures of maintaining professional excellence while honoring cultural expectations. We lift each other up and create collective strength that enables individual success.” (P20)

## 4. Discussion

This phenomenological exploration reveals the sophisticated and dynamic strategies employed by Saudi female nurses to construct integrated professional identities within mixed-gender healthcare environments. The findings fundamentally challenge simplistic conceptualizations of cultural constraints on professional development, instead revealing how cultural grounding can serve as a foundation for enhanced professional practice when appropriately understood and organizationally supported.

### 4.1. Professional Identity Formation as Cultural Integration

The study’s most significant contribution lies in demonstrating that professional identity formation among Saudi female nurses involves active integration rather than abandonment of cultural values, challenging prevailing assumptions about cultural barriers to women’s professional advancement. Our symbolic interactionist framework illuminates how participants create meaning through interactions that honor both professional requirements and cultural values, constructing hybrid identities through continuous social negotiation. Participants did not experience culture and profession as competing forces requiring difficult choices; instead, they developed sophisticated integration strategies that enhanced both cultural authenticity and professional competence [19]. This finding extends existing nursing professional identity literature by revealing how cultural values can serve as professional strengths rather than limitations when properly understood and contextually applied.

The identity integration process observed in our study reflects a key idea from symbolic interactionism: identities are not fixed traits but are actively constructed through social interactions. Participants continuously negotiated what it means to be both a “Saudi woman” and a “professional nurse” through everyday encounters with patients, families, colleagues, and societal expectations. In Mead’s terms [21], they created a unified “self” that integrates multiple “Me’s” (social roles) under the guidance of the “I” (the agentic, decision-making self). This perspective challenges essentialist views that see cultural and professional identities as inherently conflicting, instead highlighting identity as fluid, context-dependent, and continuously negotiated.

The concept of “caring as worship” emerged as a particularly powerful integration mechanism that transformed potential cultural conflicts into sources of professional motivation and spiritual fulfillment. This theological reframing enabled participants to view mixed-gender professional interactions and patient care procedures as religious obligations rather than cultural compromises, aligning with Islamic principles of serving humanity and pursuing excellence in all endeavors [22]. This finding suggests that religious and cultural frameworks, when thoughtfully applied, can provide robust foundations for professional identity development rather than constraints requiring navigation or circumvention.

Furthermore, the study reveals that successful professional identity integration requires active, ongoing negotiation rather than one-time adaptation. Participants described continuous processes of meaning-making, boundary adjustment, and strategy refinement as they encountered new professional challenges and cultural expectations. This dynamic understanding of identity formation aligns with symbolic interactionist theory’s emphasis on continuous meaning construction through social interaction, while extending its application to cross-cultural professional contexts [23].

### 4.2. Gender Dynamics and Professional Authority Development

The study illuminates complex gender dynamics that require sophisticated navigation strategies for establishing professional credibility and authority within mixed-gender healthcare teams. Participants demonstrated remarkable resilience in overcoming initial uncertainty from male colleagues through consistent demonstration of clinical excellence, strategic communication, and persistent professionalism [24]. This finding aligns with broader research on women’s leadership development in healthcare while adding important cultural dimensions specific to Islamic contexts and rapid social transformation periods.

The progression from initial doubt to professional respect, consistently reported across participants, suggests that competency-based approaches to professional development may be particularly effective in culturally diverse healthcare environments. Rather than attempting to change cultural attitudes directly, participants focused on demonstrating superior clinical skills, developing specialized expertise, and consistently delivering excellent patient outcomes. This strategy proved effective for establishing professional credibility while maintaining cultural authenticity, suggesting that excellence-based approaches may transcend cultural barriers more effectively than confrontational or assimilationist strategies.

The study also reveals how gender dynamics intersect with cultural expectations to create unique challenges requiring specialized coping strategies. Nighttime shifts, emergency situations, and intimate patient care procedures presented particular navigation challenges that participants addressed through systematic boundary management, documentation practices, and colleague education approaches. These findings provide practical guidance for healthcare organizations seeking to support female nurses from traditional cultural backgrounds while maintaining effective mixed-gender healthcare teams.

### 4.3. Organizational Support as Cultural Competency

Healthcare organizations emerged as critical facilitators or barriers to successful cultural and professional integration, with organizational cultural competency significantly influencing nurse success and retention. Organizations that provided culturally sensitive policies, flexible scheduling, professional development opportunities, and inclusive environments enabled Saudi female nurses to thrive professionally while maintaining cultural identity [12]. This finding extends organizational behavior literature by demonstrating how cultural competency at institutional levels directly impacts individual professional development outcomes.

Specifically, effective organizational practices included accommodation of religious obligations through flexible scheduling and prayer facilities, provision of culturally appropriate uniforms and workspace modifications, implementation of professional development programs that acknowledged cultural contexts while building leadership skills, and creation of inclusive policies that valued cultural diversity as organizational strength rather than accommodation burden. These practices suggest that successful cultural integration requires proactive organizational commitment rather than individual adaptation expectations.

The study reveals that organizational cultural competency extends beyond policy implementation to encompass leadership attitudes, colleague training, and environmental culture creation. Participants thrived in organizations where cultural diversity was celebrated and leveraged for enhanced patient care, while struggling in environments that viewed cultural accommodation as burden or special treatment requirement. This finding has important implications for healthcare organizations operating in culturally diverse contexts or seeking to attract and retain nurses from traditional cultural backgrounds.

### 4.4. Theoretical and Methodological Contributions

Theoretically, the study extends symbolic interactionism by demonstrating how professional identity formation occurs through cultural negotiation and integration rather than cultural transcendence or abandonment. Participants created meaning through interactions that honored both professional requirements and cultural values, revealing that identity formation in collectivistic cultures may follow integration patterns rather than the individualistic development models predominant in Western literature [25].

The study also contributes methodologically by demonstrating phenomenology’s effectiveness for exploring complex cultural and professional identity negotiations. The phenomenological approach enabled a deep understanding of lived experiences while maintaining sensitivity to cultural nuances that might be overlooked through other methodological approaches. This methodological contribution is particularly important for nursing research in culturally diverse contexts where Western theoretical frameworks may inadequately capture non-Western experiences.

### 4.5. Implications for Global Healthcare Workforce Development

The study’s implications extend beyond Saudi contexts to inform global healthcare workforce development strategies in increasingly diverse healthcare systems. As healthcare organizations worldwide recruit nurses from diverse cultural backgrounds, understanding how cultural identity intersects with professional development becomes crucial for effective workforce management and patient care quality maintenance [26], suggesting that healthcare organizations should approach cultural diversity as professional asset rather than accommodation challenge, developing policies and practices that leverage cultural knowledge for enhanced patient care while supporting professional growth. This approach requires moving beyond tolerance-based diversity frameworks toward integration-based models that recognize cultural competency as professional strength and organizational competitive advantage.

### 4.6. Study Limitations and Future Research Directions

Several limitations should be acknowledged. First, the study is culturally and geographically limited to Riyadh, Saudi Arabia. While findings may resonate with nurses in other urban Saudi or modernizing Islamic contexts, transferability to rural areas, different Islamic societies, or non-Islamic collectivist settings requires verification. Thick description is provided, but overgeneralization should be avoided.

Second, the cross-sectional design captures experiences at a single time point, limiting insight into identity development over careers; longitudinal research is needed.

Third, although the sample was diverse, it may not reflect nurses who left the profession, work in rural areas, or exclusively female units.

Fourth, interviewer characteristics (male Saudi nurse) may have influenced reporting on sensitive topics, such as gender-based challenges.

Fifth, despite rigorous translation procedures, some cultural and linguistic nuances may have been lost in moving from Arabic to English.

Future research should explore longitudinal identity development, cross-cultural comparisons in Islamic societies, male colleagues’ perspectives, patient outcomes in culturally integrated teams, and the effects of mentorship or policy interventions. Intersectional analyses considering socioeconomic status, region, and education could further clarify how gender and culture shape professional identity.

## 5. Conclusions

This phenomenological study challenges assumptions about cultural constraints on Saudi female nurses’ professional development, showing how they creatively integrate cultural authenticity with professional excellence. Rather than facing irreconcilable conflicts, participants developed synthesis strategies that enhanced both identities.

Findings highlight that effective professional identity formation in culturally diverse contexts requires moving beyond the culture–profession binary, recognizing cultural values as professional strengths, and fostering organizational systems that accommodate cultural needs while promoting growth. Saudi female nurses emerge as cultural bridges whose grounding enhances empathy, cultural competence, and patient rapport.

Healthcare organizations can support such integration by adopting comprehensive cultural competency policies, culturally informed professional development, inclusive workplace practices, and mentorship programs.

This study contributes to global nursing knowledge by demonstrating that cultural diversity and professional competence are complementary, offering insights to build inclusive healthcare environments that strengthen both patient care and workforce sustainability. While findings are grounded in Riyadh’s specific context, the core insight—that professional identity can be strengthened rather than compromised by cultural grounding when organizational support is present—may inform healthcare workforce development in other Islamic societies, immigrant nurse populations in Western contexts, and any culturally diverse healthcare setting.

## Figures and Tables

**Table 1 healthcare-13-03042-t001:** Participant Demographics and Clinical Characteristics.

Characteristic	Category	n (%)	Mean (SD)
Age	24–30 years	8 (40%)	32.1 (6.3)
	31–40 years	9 (45%)	
	41–45 years	3 (15%)	
Education	Diploma	2 (10%)	
	Bachelor’s	12 (60%)	
	Master’s	6 (30%)	
Experience	2–5 years	7 (35%)	7.4 (4.2)
	6–10 years	8 (40%)	
	11–18 years	5 (25%)	
Marital Status	Single	6 (30%)	
	Married	14 (70%)	
Clinical Specialty	Medical-Surgical	8 (40%)	
	Critical Care	5 (25%)	
	Emergency	4 (20%)	
	Pediatrics	3 (15%)	

**Table 2 healthcare-13-03042-t002:** Themes and Subthemes with Representative Quotes.

Theme	Subthemes	Representative Quote
Reconciling Traditional Values with Professional Duties	Cultural value preservation; Professional duty integration	“Working in a mixed environment doesn’t mean abandoning who I am as a Saudi woman” (P1)
Negotiating Gender Dynamics in Clinical Practice	Professional boundary management; Patient care across gender lines	“Working with male colleagues requires clear professional boundaries” (P6)
Developing Professional Identity Amid Cultural Tensions	Identity construction process; Role model development	“I’m not choosing between identities; I’m creating a new integrated identity” (P13)
Organizational Support and Environmental Adaptation	Organizational policy impact; Colleague relationships and support	“When organizations demonstrate cultural understanding, we feel valued and perform better” (P16)

**Table 3 healthcare-13-03042-t003:** Traceability from significant statements to final themes (Colaizzi’s Steps 2–5).

Significant Statement (Step 2)	Formulated Meaning (Step 3)	Theme Cluster (Step 4)	Final Theme (Step 5)
“Working in a mixed environment doesn’t mean abandoning who I am as a Saudi woman. I maintain my Islamic values, I observe proper hijab, I interact with male colleagues professionally and respectfully” (P1)	Nurse maintains cultural and religious identity markers (hijab, Islamic values) while adapting interpersonal behavior (professional respect) to mixed-gender context	Cultural value preservation within professional contexts	Theme 1: Reconciling Traditional Values with Professional Duties
“When I care for patients, I see it as an act of worship—serving humanity as Allah commanded. The Quran teaches that saving one life is like saving all humanity” (P3)	Nurse reframes clinical work through Islamic theological lens, transforming potentially conflicting activities (mixed-gender care) into religious obligations (worship, Quranic duty)	Professional duty integration through religious framework	Theme 1: Reconciling Traditional Values with Professional Duties
“Some male healthcare providers initially questioned my clinical judgment, assuming I was less capable because I’m a Saudi woman wearing hijab. I had to prove myself repeatedly through clinical excellence” (P7)	Nurse encounters gender-based competency assumptions that require ongoing demonstration of expertise to establish professional credibility	Professional credibility establishment	Theme 2: Negotiating Gender Dynamics in Clinical Practice
“I’m not choosing between identities; I’m creating a new integrated identity that’s both authentically Saudi and professionally excellent” (P13)	Nurse rejects binary identity framework (cultural OR professional) in favor of synthesis model that combines elements into unified self-concept	Identity construction as integration rather than choice	Theme 3: Developing Professional Identity Amid Cultural Tensions

## Data Availability

The qualitative datasets analysed during the current study are not publicly available due to confidentiality agreements with participants but are available from the corresponding author upon reasonable request.

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
