# Peer review of "Navigating Professional Identity and Cultural Expectations: A Phenomenological Study of Female Saudi Nurses’ Experiences in Mixed-Gender Healthcare Settings"

_healthcare, 2025, doi:10.3390/healthcare13233042_

Round 1
Reviewer 1 Report
Comments and Suggestions for Authors
Summary
Your manuscript reports a descriptive phenomenological study (Colaizzi’s seven-step analysis) of 20 Saudi nurses in mixed-gender settings, yielding four thematic domains on values, gender dynamics, identity, and organizational support.
Major issues to address
- Reporting (follow COREQ completely).
Key COREQ items are underreported: researcher characteristics/positionality, prior relationships with participants, interview setting and presence of third parties, and provision of the interview guide or prompts.
Please expand “Methods” to cover team roles/training, reflexivity, sampling pathway/recruitment, and a clear audit trail linking data to themes.
Provide a COREQ checklist as supplementary material and add an example table showing progression from significant statements → formulated meanings → clusters → - Epistemological alignment (framework vs. method).
You ground the study insymbolic interactionism yet claim Husserlian descriptive phenomenology with Colaizzi analysis; this mix needs an explicit rationale describing how an interactionist, constructionist lens coheres with an essentialist descriptive analysis and how it shaped questions, coding and theme construction.
Please justify the choice of descriptive (rather than interpretive) phenomenology under an interactionist framework, or adjust language/method to ensure philosophical congruence. - Sampling adequacy and saturation language.
Replace “theoretical saturation” with an approach appropriate to phenomenology (e.g.,information power) and explicitly document when/how adequacy was reached relative to aim, sample specificity, dialogue quality, and analytic strategy.
Clarify inclusion criteria, recruitment channels, and refusals/attrition to support transparency - Cross-language procedures (Arabic↔English).
Because interviews were conducted in Arabic, detail transcription (verbatim by whom), translation workflow (forward/back-translation, timing), and how idiomatic/religious expressions were handled to preserve meaning; if possible, present dual-language quotes.
Anchor these procedures to published guidance on translation in qualitative health research (e.g., van Nes et al., and subsequent methodological elaborations).
- Trustworthiness and verification.
You list member checking, peer review, and an audit trail, but the implementation is vague (who was checked, what was returned—transcripts, summaries, or themes—and when).
Report credibility, dependability, confirmability, and transferability strategies explicitly per Lincoln & Guba, avoiding formulaic mentions of member checking by specifying concrete procedures and artifacts
- Analytic transparency.
Include a schematic or table that traces at least two exemplar quotations through Colaizzi’s steps to the final theme, and state how disagreements were resolved beyond “consensus.”
Cite a methodological source when describing the seven steps and ensure your steps map 1:1 to that source. - Results density and claims.
Themes are plausible but occasionally assertive relative to displayed evidence; strengthen by adding more representative excerpts (including counter-examples) and clarifying how theme boundaries were decided.
Where implications extend beyond Riyadh, tie claims to data or re-frame as propositions for future study.
Minor issues
- Formatting and copy-editing.Remove “x FOR PEER REVIEW” stamps, fix broken words (“se ings”, etc.), and clean the abbreviations list (irrelevant items like “TLA”, “LD”).
- Ethics/data availability.Ethics reporting is adequate, but state whether de-identified excerpts or a codebook are available on request to enhance auditability.
- The background is current and regionally relevant; ensure all Vision-2030 and workforce claims are anchored to peer-reviewed sources already cited.
Actionable checklist (for revision letter)
- Add a COREQ-compliant methods supplement (researcher reflexivity, recruitment, setting, guide, audit trail).
- Provide a rigorous account of Arabic→English handling (who, how, when; consider dual-language quotes).
- Reconcile or re-specify the epistemology (interactionism vs descriptive phenomenology) and align text accordingly.
- Reframe sample adequacy viainformation power and document evidentiary sufficiency.
- Expand trustworthiness with concrete procedures and artifacts; de-emphasize generic “member checking.”
- Insert an analysis traceability table mapping quotes→meanings→clusters→themes per Colaizzi.
Recommendation: Major revision. Addressing the alignment, reporting, translation, verification, and analytic transparency issues will substantially strengthen credibility and transferability without changing your core contribution.

Author Response
Thank you for the opportunity to revise our manuscript, "Navigating Professional Identity and Cultural Expectations: A Phenomenological Study of Female Saudi Nurses' Experiences in Mixed-Gender Healthcare Settings." We greatly appreciate the reviewer's thoughtful and constructive feedback, which has helped us strengthen the manuscript. In the attached letter, we provide detailed responses to each comment and outline the revisions made. Thank you

Reviewer 2 Report
Comments and Suggestions for Authors
This manuscript offers a comprehensive and insightful exploration of how Saudi female nurses navigate professional identity formation within culturally sensitive, mixed-gender healthcare environments.
The literature review provides a solid foundation but could be expanded with more recent or specific studies on cultural identity in healthcare, especially in Islamic contexts.The theoretical framing with symbolic interactionism is appropriate; however, explicitly stating how this theory informs specific research questions or data interpretation would strengthen the conceptual clarity.
Regarding the methodology, the participant characteristics are well described, but more detail on recruitment procedures (e.g., how purposive sampling was conducted) and potential biases would enhance transparency.
It needs to clarify whether interviews were conducted in Arabic, translated into English for analysis, and if so, how translation fidelity was maintained.
The use of member checking is mentioned; elaborating on how this was implemented (e.g., participant feedback, adjustments made) would add rigor.
In the discussion, themes are well articulated and supported by quotes. However, some sections could benefit from more critical analysis, linking findings to existing literature more explicitly.
Finally, the practical implications have been well addressed, but suggestions could be more specific, such as proposing specific organizational policies or training interventions that explicitly support cultural integration.
Comments on the Quality of English LanguageA thorough linguistic revision is recommended to improve readability, as there are some linguistic problems in the manuscript that make some concepts unclear.
Author Response
Thank you for the reviewer’s constructive feedback. We have revised the manuscript to: expand the literature review; clarify how symbolic interactionism informs our research; provide more detail on methodology, translation, and member checking; strengthen discussion links to literature; specify practical implications; and improve English language clarity. We believe these changes address all comments and improve the manuscript.
Sincerely,
Round 2
Reviewer 1 Report
Comments and Suggestions for Authors
ok